# NARCISSUS: LEVERAGING EARLY TRAINING DYNAMICS FOR UNSUPERVISED ANOMALY DETECTION

## ABSTRACT

Anomaly detection is a critical learning task with many significant and diverse applications. Currently, semi-supervised methods provide the state-of-the-art accuracy performance but require labeled normal data for training. Unsupervised approaches, on the other hand, do not have this requirement but can only offer inferior anomaly detection performance. In this paper, we introduce NARCISSUS, a novel unsupervised anomaly detection method that achieves accuracy comparable to semi-supervised approaches. Our key insight is that a learning model when training with a mix of normal and sparse anomalous data converges first on normal data. Leveraging this insight, NARCISSUS employs a tailored early stopping scheme, eliminating the need for pseudo labels and costly label generation interactions. It also offers systematic solutions to minimize the influence of model uncertainty, ensuring robust detection. NARCISSUS is model-agnostic and can therefore make use of even a semi-supervised anomaly detection model underneath, thereby turning it into an unsupervised one. Comprehensive evaluations using time series, image and graph datasets show that NARCISSUS provides similar or better detection performance compared to best-performing semi-supervised methods while not requiring labeled data.

## 1 INTRODUCTION

Anomaly detection (AD) (Han et al., 2022; Pang et al., 2021; Chandola et al., 2009), or outlier detection, is a critical machine learning (ML) task with diverse applications, including anti-money laundering, network diagnostics, rare disease detection, and social media analysis. AD algorithms identify data instances that significantly deviate from the norm. While traditional unsupervised methods (Nakamura et al., 2020; Ahmad et al., 2017) detect anomalies without prior knowledge of normal or anomalous data, emerging (semi-)supervised approaches (Lai et al., 2024; Tuli et al., 2022) demonstrate improved accuracy by leveraging prior information.

However, (semi-)supervised approaches rely on well-labeled data, which presents two key challenges for anomaly detection: (1) *Data Availability*: Obtaining a well-labeled training dataset is often difficult, and it can be challenging to ensure the dataset is entirely anomaly-free; (2) *Overfitting*: The trained model may overfit to the training data, resulting in inaccurate detection when faced with distribution shifts. Conversely, unsupervised anomaly detection methods generally under-perform compared to semi-supervised approaches that provide state-of-the-art accuracy (Pang et al., 2021; Han et al., 2022). The absence of supervision makes it more difficult for unsupervised methods to effectively distinguish anomalies from normal patterns.

The above discussion suggests that combining unsupervised and semi-supervised learning can be a promising way forward. Indeed, previous works have attempted training with bootstrapped datasets that mix normal and abnormal data; however, a significant performance drop is consistently observed compared to training with exclusively normal data (Livernoche et al., 2024; Han et al., 2022). There are also self-supervised methods that assign pseudo labels generated by an unsupervised detection technique to train semi-supervised models (Li et al., 2021; Zhang et al., 2023). However, the effectiveness of the self-supervised approach is constrained by the convergence speed and accuracy of the initial unsupervised method, providing only modest improvements to overall accuracy. Thus, *achieving high-accuracy unsupervised anomaly detection remains a significant challenge*.

Motivated by the above, we aim to design an unsupervised anomaly detection scheme that offers the state-of-the-art accuracy. Our work is inspired by research in binary classification, where models tend to exhibit significantly lower loss on low-influence data early in the training process (Paul et al., 2021), as well as the use of early stopping as a Nonparametric Variational Inference method (Duvenaud et al., 2016). Our key insight is that when training a model for some learning task on a mix of normal and anomalous data, the model converges faster on normal data while struggling to fit to anomalous data. In other words, at a certain point in the training process, the model would have effectively learned to fit the normal data but continues to exhibit higher loss on anomalous data (see Figure 1 for an illustration). Note that this phenomenon is model-agnostic, as all models undergo a similar process when fitting the training dataset.

In this paper, we introduce NARCISSUS, a new unsupervised anomaly detection method that exploits the above insight. NARCISSUS leverages training dynamics for accurate and robust anomaly detection with unlabeled data through a combination of a tailored early stopping algorithm and an ensemble method. Early stopping enables identifying anomalies without the additional data labeling cost, while the highly parallel ensemble method mitigates the impact of epistemic uncertainty in a lightweight yet reliable way. Our comprehensive evaluation demonstrates that NARCISSUS allows successfully turning an AD algorithm originally designed for semi-supervised anomaly detection to an unsupervised setup while maintaining or often improving accuracy.

**Our main contributions are as follows**:

- **NARCISSUS**: We propose a novel unsupervised anomaly detection method that achieves high accuracy without requiring labeled data.
- **Key Insight – Training Dynamics**: We observe, analyze, and leverage the unique dynamics when performing model training with a mix of normal and sparse anomalous data, where the model consistently converges on normal data first.
- **Mitigating Model Uncertainty**: We employ an ensemble approach to reduce the impact of epistemic uncertainty, enhancing the robustness of anomaly detection.

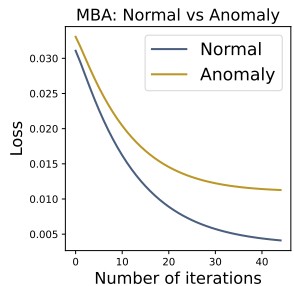

Figure 1: Evolution of loss during training of TranAD (Tuli et al., 2022) on MBA dataset (Moody & Mark, 2001). Convergence speed faster for normal data relative to anomalous data.

## 2 RELATED WORK

Anomaly detection methods mainly fall into two categories based on the availability of labeled data: (1) Unsupervised: No labels are available. These methods aim to detect anomalies solely based on the inherent properties of the data (Ruff et al., 2018; Zong et al., 2018; Nakamura et al., 2020; Zhang et al., 2023); (2) Semi-supervised: Only partial labels are known, typically for a subset of normal data (Ahmad et al., 2017; Li et al., 2019; Zhao et al., 2020; Audibert et al., 2020; Deng & Hooi, 2021; Tuli et al., 2022; Lai et al., 2024). These methods leverage the available labeled normal data to improve detection accuracy. Fully supervised scenarios, where ample labeled data for both normal and anomalous cases exist, are uncommon in real-world applications due to the rarity and diversity of anomalies. Therefore, our focus is on unsupervised and semi-supervised settings. In addition to different levels of supervision, anomaly detection have been applied across diverse data formats, including time series (Tuli et al., 2022; Zhang et al., 2023), images (Schlegl et al., 2017; Roth et al., 2022; Livernoche et al., 2024), and graphs (Zheng et al., 2019b; Ma et al., 2021).

Recent studies in binary classification and influence scores (Zhang et al., 2024; Paul et al., 2021; Liu et al., 2021) are closely related to anomaly detection, as the latter also involves classifying data into normal or anomalous categories. Paul et al. (2021) introduced the EL2N score (§A.1) to quantify a sample's contribution to a binary detection model, observing that high-value samples exhibit higher loss during the early training phase. Similarly, Liu et al. (2021) identified that classification models do not converge uniformly across categories. Although anomaly detection models are typically regression or reconstruction based rather than following traditional classification, these insights inspire us to explore similar patterns during training for anomaly detection. Other related works include influence functions (Koh & Liang, 2017) and Kalman Filters with outlier robustness (Duran-Martin

et al., 2024). While these methods can highlight outliers by estimating the loss on specific samples, they lack the expressiveness needed to capture the full complexity of anomaly detection.

# 3 PRELIMINARIES

## 3.1 OBJECTIVE OF ANOMALY DETECTION

Given a time series $\mathbb{X} = \{x_0, x_1, \ldots, x_{t-1}, x_t\}$, the anomaly detection problem for time series can be described as: learn an anomaly score $\mathbf{s}_t = \{s_{t,i}\}$ for each time stamp $t$, where $i$ represents different attributes in a multivariate time series so that there is a threshold $T_a$ (either learnt or predefined) to compute a label $l_t$ for each time stamp,

$$l_t = \begin{cases} 0, & \text{if } \forall i, s_{t,i} < T_a, \\ 1, & \text{if } \exists i, s_{t,i} \geq T_a \end{cases} \tag{1}$$

where $l_t = 1$ means there is an anomaly at time $t$. Denoting the ground truth anomaly label with $\hat{l}_t$, the objective of anomaly detection is to ensure $\forall t, l_t = \hat{l}_t$. Note that the score for a timestamp can be interpreted similarly to pixel or node scores in image and graph setups. Anomaly detection applies this concept consistently across different data formats.

## 3.2 SEMI-SUPERVISED DETECTION AND ANOMALY SCORE

The state-of-the-art approach for obtaining an anomaly score follows a semi-supervised learning paradigm, typically through the prediction or reconstruction loss of a model pretrained with normal data (Lai et al., 2024; Deng & Hooi, 2021). The rationale of this scoring method is that with a properly trained model using normal data, the high prediction (reconstruction) error for a test input indicates that the input is an outlier (Hinton & Salakhutdinov, 2006), which can be represented as

$$p(x \in \mathbb{U} | \mathcal{L}(x) > \sigma) < \epsilon \tag{2}$$

where $\mathcal{L}(\cdot)$ is the loss function, $\sigma$ is variance of errors for inputs within the training distribution, $\mathbb{U}$ is the set of normal data, $\epsilon$ is a small positive . Equation 2 is the fundamental theoretical support for all existing deep (semi-)supervised detection methods. The effectiveness of a deep detection method depends on how well the learnt function models the normal data.

Comparing equations (1) and (2), if we use the loss as an anomaly score, set $T_a$ to $k\sigma$, and classify all inputs with non-zero probability density as normal samples, they become equivalent.

# 4 NARCISSUS: ANOMALY DETECTION DURING TRAINING

## 4.1 UNSUPERVISED ANOMALY DETECTION FROM A SEMI-SUPERVISED VIEWPOINT

In unsupervised setup, we cannot leverage Equation 2 directly since we need to learn a function that describes normal data to compute $\mathcal{L}(\cdot)$. Let us consider a heuristic approach in which training data is randomly selected for semi-supervised learning. The ideal detection function $f$ should perfectly fit all normal data with negligible loss while failing to fit anomalous data. This leads us to the following optimization problem:

$$\arg\max_{\mathbb{Y} \subset \mathbb{X}, f \in \mathcal{F}_{\mathbb{U}}} \frac{\sum_{y \in \mathbb{Y}} |\mathcal{L}_f(y)|}{|\mathbb{Y}|}$$
$$\text{s.t.} \quad |\mathbb{Y}| < N_{max} \ll |\mathbb{X}| \tag{3}$$
$$|\mathcal{L}_f(y)| > \epsilon, \ \forall y \in \mathbb{Y}$$
$$E[(E(\tilde{f}) - E(f)) | \mathbb{U}] < \epsilon$$

where $\mathbb{X}$ is the set of all data, $\mathbb{Y}$ is the pseudo anomalous set, $\mathcal{L}_f(\cdot)$ computes the loss value of function $f$ on different inputs, $\mathcal{F}_{\mathbb{U}}$ is set of all possible functions trained on pseudo normal subset $\mathbb{U} = \mathbb{X} - \mathbb{Y}$, $\tilde{f}$ is the ideal function that fits the normal data, $\epsilon \ll 1$. Intuitively, it tries to find subset $\mathbb{Y}$ so that (i) $\mathbb{Y}$ is the small subset (the first constraint), $|\mathbb{Y}| \ll |\mathbb{X}|$ reflects that anomalies

are rare compared to the total dataset, and $N_{\max}/|\mathbb{X}|$ represents the expected anomaly sparsity, as statistically, only a small fraction of long-term observation data can be classified as anomalies; (ii) find $f$ that describes pseudo normal set $\mathbb{U}$ (the third constraint) but yields significant error on pseudo anomalous data (the second constraint). Because of the hypothesis in Eq. 2, the objective function can also be interpreted as the probability of $y$ belonging to normal set to be nearly zero.

**Lemma 4.1.** *If there exist anomalous point in the dataset, the optimization problem defined in Eq. 3 must admit at least one solution, though the solution may not be unique.*

*Proof.* Proof is by contradiction, see §A.2. □

The bootstrapping method used in Livernoche et al. (2024) neither maximizes the objective function in the above optimization problem nor does it satisfy the associated constraints. It simply performs a random search to resolve Eq. 3. Consequently, we can expect its performance to be inferior to the semi-supervised approach, that has more informed guidance through labeled normal data.

## 4.2 DATA CHARACTERISTICS

We identify the following common characteristics across typical anomaly detection scenarios: (i) anomalies are generally sparse, constituting only a small fraction of the entire dataset; and (ii) the data is well-bounded, with the dynamic range of anomalous points not significantly (orders of magnitude) exceeding that of normal data. Observation (i) aligns with the definition of an anomaly or outlier – data that deviates from the normal majority. Observation (ii) applies specifically to cases requiring deep learning, as simpler threshold-based methods could otherwise detect anomalies. Later, we demonstrate how these characteristics result in different convergence rates for normal and anomalous data, an insight we leverage for unsupervised anomaly detection.

## 4.3 STOCHASTIC GRADIENT DESCENT LEARNS TO FIT NORMAL DATA FIRST

**Theorem 4.2.** *Given a regression task on a dataset $\mathbb{X}$ containing sparse anomalous data $\mathbb{Y}$, and the rest is normal data $\mathbb{U}$, ($N_a = |\mathbb{Y}| \ll |\mathbb{X} - \mathbb{Y}| = |\mathbb{U}| = N_n$), suppose the gradient of the loss function with respect to the model are bounded, i.e., for any normal data $x \in \mathbb{U}$: $\nabla_\theta L(x, f_t(x))\| \leq \delta_n$, and for any anomalous data $y \in \mathbb{Y}$: $\nabla_\theta L(y, f_t(y))\| \leq \delta_a$ where $\nabla_\theta L(x, f_t(x))$ denotes the gradient of the loss with respect to the model parameter $\theta$ at iteration $t$, and $\delta_n$ and $\delta_a$ are upper bounds on the gradient norms for normal and anomalous data, respectively.*
*If the condition $N_n \cdot \delta_n \gg N_a \cdot \delta_a$ holds, then training the model $f$ using stochastic gradient descent (SGD) will result in convergence towards fitting the normal data, with a bounded difference compared to training on anomaly-free data.*

*Proof.* Consider an iteration $t$ of SGD, the update to the model parameter $\theta$ can be decomposed into contributions from normal and anomalous data:

$$\Delta_t = -\eta \left( \sum_{x \in \mathbb{X} - \mathbb{Y}} \nabla_\theta \mathcal{L}(x, f_t(x)) + \sum_{y \in \mathbb{Y}} \nabla_\theta \mathcal{L}(y, f_t(y)) \right)$$

where $\eta$ is the learning rate. By assumption, the gradients are bounded, hence the total gradient contribution bounds are: Normal data contribution:

$$| \sum_{x \in \mathbb{X} - \mathbb{Y}} \nabla_\theta \mathcal{L}(x, f_t(x))| \leq N_n \cdot \delta_n$$

Anomalous data contribution:

$$| \sum_{y \in \mathbb{Y}} \nabla_\theta \mathcal{L}(y, f_t(y))| \leq N_a \cdot \delta_a$$

To ensure that the model converges towards fitting the normal data, the influence of the normal data on the parameter updates must significantly outweigh that of the anomalous data. This requires:

$$N_n \cdot \delta_n \gg N_a \cdot \delta_a$$

This condition implies that the cumulative gradient magnitude from normal data is much larger than that from anomalous data. Since $N_a \ll N_n$, and assuming $\delta_a$ is not excessively larger than $\delta_n$, the anomalous data contributes relatively little to the overall gradient. Specifically, the ratio of the total anomalous gradient contribution to the normal gradient contribution satisfies:

$$\frac{N_a \cdot \delta_a}{N_n \cdot \delta_n} \ll 1$$

Under the above condition, the parameter updates are primarily influenced by the normal data. The anomalous data introduce a bounded perturbation, which can be considered as noise in the optimization process. According to the convergence properties of SGD with bounded noise, the model will still converge towards the optimal parameters for the normal data, possibly at a slower rate or with a small bias. $\qquad\square$

To ensure the model converge towards to normal data, according to Theorem 4.2, we need ensure

$$|\sum_{N_n} \nabla \mathcal{L}(x, f_t(x))| \gg N_a \cdot \delta_a$$

Apparently, we also have

$$N_n \cdot \delta_n \geq \sum_{N_n} |\nabla \mathcal{L}(x, f_t(x))| \geq |\sum_{N_n} \nabla \mathcal{L}(x, f_t(x))|$$

If the loss on normal dataset is bounded by $\delta_n$, $|\nabla \mathcal{L}(x, f_t(x))/N_n| \leq \delta_n$. If the $N_n \cdot \delta_n \gg N_a \cdot \delta_a$, *i.e.*, $\frac{\delta_a}{\delta_n} \ll \frac{N_n}{N_a}$, then the model will very likely converge to fit the normal data. $\frac{N_n}{N_a}$ is a constant large value, therefore the ratio $\frac{\delta_a}{\delta_n}$ can reflects whether the model will converge on normal data. Figure 1 illustrates this empirically, showing that the model converges on normal data significantly faster than anomalous data.

Notice that in binary classification problems, the samples that contribute most to the training can often be identified early in the process (Paul et al., 2021). In the Appendix A.1, Theorem A.2, we clarify the fundamental differences between general binary classification and anomaly detection through theoretical analysis and experiments. We demonstrate that the concepts and methods proposed in Paul et al. (2021); Zhang et al. (2024) are not applicable to anomaly detection due to the fundamental differences in training approaches.

## 4.4 VERY EARLY STOPPING

Theorem 4.2 suggests an important corollary:

**Corollary 4.3.** *The first converged data are more likely to be normal and the model starts to learn anomalous data only when the loss on normal data is small.*

Inspired by this, we propose **V**ery **E**arly **S**topping (VES), a tailored early stopping scheme for training the model according to selected validation sets. The key idea behind VES is that the model begins by learning patterns from the normal data and only starts to fit anomalous data when the loss on the normal data has become sufficiently small. The complete process is enclosed in §A.3, Algorithm 3.

---

**Algorithm 1** (**Core Algorithm**) Very Early Stopping for Unsupervised Anomaly Detection

---

1: **if** $N \leq N_{\max}$ and $\mathcal{E}_N$ have not converge **then** $\qquad\triangleright N_{\max}$ is the maximum epoch number

2: $\quad$ **while** $i < \frac{|\mathbb{T}'|}{|\tau_i|}$ **do**

3: $\qquad v_i \leftarrow \frac{\sum_{\forall t \in \tau_i} \mathcal{L}(t, f(t))}{|\tau_i|}$

4: $\qquad i \leftarrow i + 1$

5: $\quad \mathbb{V}' \leftarrow \{v_i\}, v_i < q_{\mathrm{mean}}(\mathbb{V}, \eta\%) \qquad\qquad\triangleright$ Filter out top $\eta\%$ loss on validations $\{\tau\}$ by mean

6: $\quad \mathbb{V}^* \leftarrow \{v_i\}, v_i < q_{\max}(\mathbb{V}, \eta\%) \qquad\triangleright$ Filter out top $\eta\%$ loss on validations $\{\tau\}$ by maximum

7: $\quad \mathcal{E}_N \leftarrow \frac{\sum \mathbb{V}' \cap \mathbb{V}^*}{|\mathbb{V}' \cap \mathbb{V}^*|} \qquad\qquad\qquad\qquad\qquad\qquad\qquad\triangleright$ Compute the constraint

---

Alg. 1 illustrates the core component of VES. Specifically, we randomly select $\frac{|\mathbb{T}'|}{|\tau_i|}$ small validation subsets $\tau_i$ from the original dataset $\mathbb{X}$ to get the large validation set $\mathbb{T}'$. After each epoch, the loss at each timestamp within every validation subset is computed. The validation subsets are then sorted based on their mean and maximum loss values. We filter out the top $\eta\%$ of validation subsets with the highest mean losses and separately filter out the top $\eta\%$ with the highest maximum losses. Statistically, if we ignore the top $\eta\%$ high loss on validation set, then the rest part will mostly reflect the convergence on normal data, where the $\eta\%$ is the upper bound of the portion of anomalous data. Therefore, we obtain the following Alg. 1 to identify if the model converges on normal data. More detailed discussion in §A.4.

The intersection of the remaining subsets $\mathbb{V}' \cap \mathbb{V}^*$ is used to calculate the convergence metric, $\mathbb{V}$ represents the loss of all validation set, $\mathbb{V}'$ and $\mathbb{V}'$ represent the loss of selected subset in line 5 and 6, Alg.1. Once the the model converges on the intersection (conventional early stopping is applied), we deem that the model has converged on normal data and constraint $E[(E(\tilde{f}) - E(f))|\mathbb{U}] < \epsilon$ in Eq. 3 is met.

By stopping the training early, we prevent the model from starting to fit the anomalous data, which could lead to over-fitting and reduced anomaly detection performance. VES thus enhances the model's ability to generalize to unseen normal data and to better distinguish anomalies during inference.

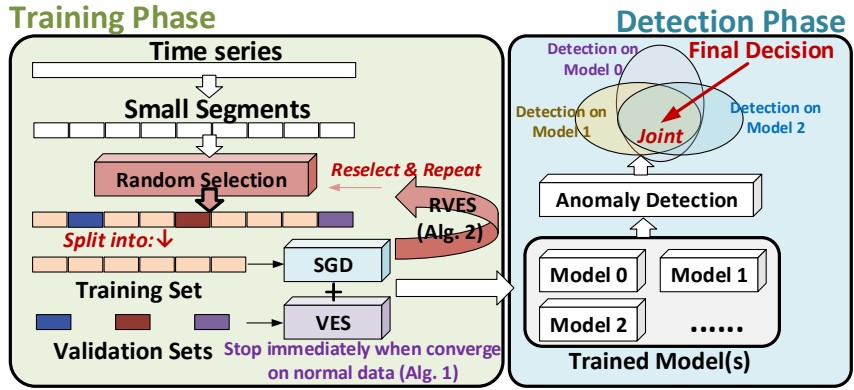

Figure 2: Main process of NARCISSUS. Using RVES in Alg. 2 randomly select validation sets and retrain the same model as different models in an ensemble, and take the joint set in the ensemble for the final decision.

---

**Algorithm 2** Robust VES by repeating

---

**Require:** $|\mathbb{T}'_i| = |\mathbb{T}'_j|; |\mathbb{T}'_i \cap \mathbb{T}'_j| = 0, \forall i \neq j; \bigcap_i \mathbb{T}'_i = \mathbb{X}$
**Ensure:** $E[(E(\tilde{f}) - E(f))|\mathbb{U}] < \epsilon, \forall \mathbb{T}'_i$
1: Random Initialization $\mathbb{M} = \{M_i\}$      ▷ Non-overlap random mask with window size $|\tau_0|$.
2: $\mathbb{T}'_i \leftarrow \mathbb{T} \cdot M_i$              ▷ Refer to Figure 2
3: Initialize **VES**$(\cdot)$            ▷ Initialize Alg. 1
4: $N = 0$
5: **while** $N < |\mathbb{M}|$ **do**
6:    $\mathbb{A}_N \leftarrow$ **VES**$(\mathbb{T}'_N)$       ▷ Record the detection result with different $\mathbb{T}'_i$
7:    $N = N + 1$
8: $\mathbb{A} = \bigcap \mathbb{A}_i, \forall i$       ▷ Combine (joint set) the result of different validation set.

---

Note that in Alg. 1, empirically we can choose a large $\eta$ to ensure most of the considered validation data are normal, which will accelerate the VES process. We include more optimisations to further accelerate VES in Appendix A.5.

Building on VES, we propose the framework of NARCISSUS, which is illustrated in Figure 2. NARCISSUS improves overall robustness through Robust VES (RVES) in Alg. 2. As the red arrow of

Figure 2 illustrates, RVES enhances robustness by repeatedly performing VES with random variations and leveraging the trained model in different iterations for ensemble learning.

Given the limited size of the dataset, it is challenging to avoid stochastic biases when selecting the validation set. The scarcity of training data further amplifies model uncertainty, and the validation set selection exacerbates this issue by reducing the data available for training. To mitigate these challenges, RVES in Alg. 2 adopts the following strategies:

- Try multiple random validation sets until a significant portion of the dataset is covered. This reduces the impact of selecting a biased validation set.
- Training with different datasets in RVES jointly performs **random initialization** (Lee et al., 2015; Lakshminarayanan et al., 2017), and **Nonparametric Variational Inference** (Duvenaud et al., 2016), which can reflect the epistemic uncertainty and enhance the robustness of detection.

Informally, NARCISSUS includes a while-loop of VES, with VES serving as the core component, while RVES reflects the ensemble.

## 5 EVALUATION

### 5.1 METHODOLOGY

We consider three kinds of baselines, including: (i) unsupervised anomaly detection methods, (ii) semi-supervised methods with unsupervised bootstrapping (Han et al., 2022; Livernoche et al., 2024), and (iii) semi-supervised methods in their original setup. Baseline type (i) refers to the methods that perform anomaly detection in an unsupervised manner, such as Zong et al. (2018). Type (ii) is the most common approach for applying a semi-supervised model to an unsupervised scenario, using randomly selected data as normal data for bootstrapping during training. The difference between type (ii) and (iii) is the training dataset, where (iii) is trained with clean dataset free of anomalies.

Self-supervised methods (Zhang et al., 2023) with pseudo labels and interactive training are not considered because they would need a method like NARCISSUS as a module in the self-supervised workflow. Since NARCISSUS significantly outperforms existing unsupervised detection methods, it will inherently boost performance in self-supervised setups. Empirically, we find that models trained with NARCISSUS match or surpass semi-supervised models with comparable computational overhead, making self-supervised workflows unnecessary due to their limited improvement and higher computational cost.

We evaluate different AD methods considering widely used metrics: F1 score, precision (P), recall (R), and AUC – area under receiver operating characteristic (ROC) curve.

### 5.2 MULTI-VARIATE TIME SERIES ANOMALY DETECTION

For multi-variate time series (MTS), as a base model in NARCISSUS, we mainly consider the following semi-supervised methods: LSTM-NDT (Hundman et al., 2018a), OmniAnomaly (Su et al., 2019a), USAD (Audibert et al., 2020), MTAD-GAT (Zhao et al., 2020), GDN (Deng & Hooi, 2021), TranAD (Tuli et al., 2022), and NPSR (Lai et al., 2024). Besides, we also consider common methods that are designed solely for unsupervised detection – DAGMM (Zong et al., 2018), MSCRED (Zhang et al., 2019) and Merlin (Nakamura et al., 2020). We selected these baselines because they are highly representative, consistently deliver state-of-the-art performance within their respective method categories, and have been widely reproduced and validated across numerous studies (Lai et al., 2024; Tuli et al., 2022; Han et al., 2022).

We evaluate NARCISSUS on datasets that are widely used in previous works (Lai et al., 2024; Tuli et al., 2022) with the same prepossessing methods, namely: NAB (Numenta Anomaly Benchmark) (Ahmad et al., 2017), SMD (Server Machine Dataset) (Su et al., 2019b), MBA (MIT-BIH Supraventricular Arrhythmia Database) (Moody & Mark, 2001), SMAP (Soil Moisture Active Passive) (Hundman et al., 2018b), SWaT (Secure Water Treatment) (Goh et al., 2017), and a Synthetic dataset used in Tuli et al. (2022).

The comparison between NARCISSUS and conventional unsupervised learning methods are demonstrated in Table 1. Overall, the implementation of NARCISSUS with different base semi-supervised

| Method | NAB | | | MBA | | | SMD | | |
|---|---|---|---|---|---|---|---|---|---|
| | P | AUC | F1 | P | AUC | F1 | P | AUC | F1 |
| DAGMM | 0.7622 | 0.7272 | 0.7443 | 0.9103 | 0.9954 | 0.9491 | 0.7453 | 0.9987 | 0.6890 |
| MSCRED | 0.8522 | 0.7606 | 0.7502 | 0.7276 | 0.9921 | 0.8414 | 0.9116 | 0.9842 | 0.9437 |
| MERLIN | 0.8013 | 0.7262 | 0.8414 | 0.2871 | 0.7158 | 0.3842 | 0.7619 | 0.7542 | 0.8018 |
| N-LSTM-NDT | 0.6400 | 0.6667 | 0.8374 | 0.9736 | 0.9671 | 0.9042 | 0.7578 | 0.8294 | 0.9152 |
| N-OmniAnomaly | 0.8421 | 0.6667 | 0.8754 | 0.8881 | 0.9946 | 0.9401 | 0.8344 | 0.9716 | 0.8196 |
| N-USAD | 0.8571 | 0.9995 | 0.9231 | 0.8453 | 0.9531 | 0.9287 | 0.9110 | 0.9921 | 0.9235 |
| N-MTAD-GAT | 0.9999 | 0.6667 | 0.5000 | 0.8670 | 0.9607 | 0.9220 | 0.9990 | 0.8635 | 0.8416 |
| N-GDN | 0.8889 | 0.9996 | 0.9412 | 0.8598 | 0.9583 | 0.9246 | 0.7980 | 0.9872 | 0.8350 |
| N-TranAD | 0.8889 | 0.9996 | 0.9412 | 0.9461 | 0.9854 | 0.9723 | 0.9996 | 0.9220 | 0.9152 |
| N-NPSR | 0.8571 | 0.9995 | 0.9231 | 0.8595 | 0.9582 | 0.9244 | 0.8117 | 0.9867 | 0.8950 |

| Method | SMAP | | | SWaT | | | Synthetic | | |
|---|---|---|---|---|---|---|---|---|---|
| | P | AUC | F1 | P | AUC | F1 | P | AUC | F1 |
| DAGMM | 0.8523 | 0.7326 | 0.8602 | 0.7778 | 0.9519 | 0.6586 | 0.9543 | 0.9988 | 0.9766 |
| MSCRED | 0.8130 | 0.9149 | 0.9485 | 0.9999 | 0.8376 | 0.6879 | 0.9776 | 0.9994 | 0.9887 |
| MERLIN | 0.1577 | 0.9999 | 0.7426 | 0.6560 | 0.7140 | 0.5022 | 0.8543 | 0.7576 | 0.8332 |
| N-LSTM-NDT | 0.8060 | 0.9891 | 0.9885 | 0.9833 | 0.8436 | 0.9997 | 0.9231 | 0.9988 | 0.9231 |
| N-OmniAnomaly | 0.8175 | 0.9216 | 0.9218 | 0.9992 | 0.9998 | 0.9887 | 0.9776 | 0.9994 | 0.9887 |
| N-USAD | 0.8455 | 0.9692 | 0.9066 | 0.9977 | 0.8438 | 0.8143 | 0.9562 | 0.9988 | 0.9776 |
| N-MTAD-GAT | 0.8485 | 0.9806 | 0.9180 | 0.9700 | 0.8462 | 0.8101 | 0.9449 | 0.9985 | 0.9717 |
| N-GDN | 0.9440 | 0.9823 | 0.9603 | 0.9762 | 0.8497 | 0.8166 | 0.9658 | 0.9991 | 0.9826 |
| N-TranAD | 0.8503 | 0.9809 | 0.9191 | 0.9933 | 0.8436 | 0.8128 | 0.9776 | 0.9994 | 0.9887 |
| N-NPSR | 0.9401 | 0.9820 | 0.9587 | 0.9977 | 0.8438 | 0.8143 | 0.9856 | 0.9996 | 0.9928 |

Table 1: Performance comparison of unsupervised AD methods with semi-supervised methods trained with NARCISSUS (named as "**N-[Method Name]**") in terms of Precision (P), AUC and F1 score metrics on multiple different datasets.

| Methods | NAB | | | | | | MBA | | | | | |
|---|---|---|---|---|---|---|---|---|---|---|---|---|
| | Semi-Supervised | | | NARCISSUS | | | Semi-Supervised | | | NARCISSUS | | |
| | P | AUC | F1 | P | AUC | F1 | P | AUC | F1 | P | AUC | F1 |
| USAD | 0.8421 | 0.8330 | 0.7442 | 0.8571 | 0.9995 | 0.9231 | 0.8953 | 0.9701 | 0.9443 | 0.8453 | 0.9531 | 0.9287 |
| MTAD-GAT | 0.8421 | 0.8478 | 0.7752 | 0.9999 | 0.6667 | 0.5000 | 0.8390 | 0.9551 | 0.9124 | 0.8670 | 0.9607 | 0.9220 |
| GDN | 0.8129 | 0.8542 | 0.7998 | 0.8889 | 0.9996 | 0.9412 | 0.8832 | 0.9528 | 0.9332 | 0.8598 | 0.9583 | 0.9246 |
| NPSR | 0.4615 | 0.9965 | 0.6316 | 0.8571 | 0.9995 | 0.9231 | 0.8578 | 0.9576 | 0.9235 | 0.8595 | 0.9582 | 0.9244 |
| TranAD | 0.8889 | 0.9541 | 0.9364 | 0.8889 | 0.9996 | 0.9412 | 0.9569 | 0.9885 | 0.9780 | 0.9461 | 0.9854 | 0.9723 |

| Methods | SMD | | | | | | SMAP | | | | | |
|---|---|---|---|---|---|---|---|---|---|---|---|---|
| | Semi-Supervised | | | NARCISSUS | | | Semi-Supervised | | | NARCISSUS | | |
| | P | AUC | F1 | P | AUC | F1 | P | AUC | F1 | P | AUC | F1 |
| USAD | 0.9060 | 0.9933 | 0.9495 | 0.9110 | 0.9921 | 0.9235 | 0.8139 | 0.9890 | 0.8974 | 0.8455 | 0.9692 | 0.9066 |
| MTAD-GAT | 0.8210 | 0.9921 | 0.8683 | 0.9990 | 0.8635 | 0.8416 | 0.7518 | 0.9841 | 0.8583 | 0.8485 | 0.9806 | 0.9180 |
| GDN | 0.7170 | 0.9924 | 0.8342 | 0.7980 | 0.9872 | 0.8350 | 0.8293 | 0.9901 | 0.9067 | 0.9440 | 0.9823 | 0.9603 |
| NPSR | 0.8110 | 0.9689 | 0.8843 | 0.8117 | 0.9867 | 0.8950 | 0.9236 | 0.9798 | 0.9496 | 0.9401 | 0.9820 | 0.9587 |
| TranAD | 0.9262 | 0.9974 | 0.9605 | 0.9996 | 0.9220 | 0.9152 | 0.8175 | 0.9892 | 0.8996 | 0.8503 | 0.9809 | 0.9191 |

| Methods | SWaT | | | | | | Synthetic | | | | | |
|---|---|---|---|---|---|---|---|---|---|---|---|---|
| | Semi-Supervised | | | NARCISSUS | | | Semi-Supervised | | | NARCISSUS | | |
| | P | AUC | F1 | P | AUC | F1 | P | AUC | F1 | P | AUC | F1 |
| USAD | 0.9977 | 0.8460 | 0.8143 | 0.9977 | 0.8438 | 0.8143 | 0.9619 | 0.9990 | 0.9806 | 0.9562 | 0.9988 | 0.9776 |
| MTAD-GAT | 0.9718 | 0.8464 | 0.8109 | 0.9700 | 0.8462 | 0.8101 | 0.9600 | 0.9989 | 0.9796 | 0.9449 | 0.9985 | 0.9717 |
| GDN | 0.9697 | 0.8462 | 0.8101 | 0.9762 | 0.8497 | 0.8166 | 0.9677 | 0.9916 | 0.9836 | 0.9658 | 0.9991 | 0.9826 |
| NPSR | 0.9697 | 0.8462 | 0.8101 | 0.9977 | 0.8438 | 0.8143 | 0.9677 | 0.9916 | 0.9836 | 0.9856 | 0.9996 | 0.9928 |
| TranAD | 0.9760 | 0.8491 | 0.8151 | 0.9933 | 0.8436 | 0.8128 | 0.9091 | 0.9975 | 0.9524 | 0.9776 | 0.9994 | 0.9887 |

Table 2: Performance comparison of various semi-supervised AD methods with their original training setup ("Semi-Supervised") with same methods when trained with NARCISSUS in an unsupervised manner ("NARCISSUS") in terms of Precision (P), AUC and F1 score metrics on multiple different datasets.

methods significantly outperforms all unsupervised detection methods in every metric, including precision, AUC, and F1 score. In Table 1, we also observe that in a few specific cases, such as applying NARCISSUS to MTAD-GAT on the NAB dataset, performance is affected due to the small dataset size with only 8,000 timestamps, making it challenging to train MTAD-GAT effectively with NARCISSUS. This special case, however, does not reflect the reliability of NARCISSUS, as NARCIS-

| Metric | Semi-supervised PatchCore | | | PatchCore bootstrapping | | | PatchCore NARCISSUS | | |
|---|---|---|---|---|---|---|---|---|---|
| | 25% | 10% | 1% | 25% | 10% | 1% | 25% | 10% | 1% |
| AUC↑ | 0.9811 | 0.9810 | 0.9802 | 0.9553 | 0.9567 | 0.9549 | 0.9811 | 0.9803 | 0.9802 |
| Error↓ | 1.9 | 1.9 | 2.0 | 4.4 | 4.3 | 4.4 | 1.9 | 2.0 | 2.0 |

Table 3: Implement NARCISSUS and bootstrapping on PatchCore.

| Method | Precision↑ | Recall↑ | Sensitivity↑ | Specificity↑ | AUC↑ |
|---|---|---|---|---|---|
| AnoGAN | 0.8839 | 0.7312 | 0.7281 | 0.8929 | 0.8912 |
| w/ bootstrapping | 0.8672 | 0.7113 | 0.7196 | 0.8901 | 0.8795 |
| w/ NARCISSUS | 0.8812 | 0.7352 | 0.7312 | 0.8973 | 0.8925 |

Table 4: Implement NARCISSUS and bootstrapping on AnoGAN.

SUS with all other base methods performs similarly or better than their semi-supervised counterparts as we show next in Table 2. The low F1 score is an inherent limitation of MTAD-GAT, not of the NARCISSUS itself.

Recall that NARCISSUS introduces a novel training approach that can enable direct training of semi-supervised detection models without the need for any pseudo labels – a capability that, to the best of our knowledge, is entirely unique. We then compare NARCISSUS against semi-supervised methods, where a substantial amount of normal data is available to the baselines during the training phase. The corresponding results are showed in Table 2, where in most cases, the difference is F1 score is within 0.02. We observe many instances where the unsupervised NARCISSUS significantly outperforms semi-supervised methods, such as on the SMAP dataset, due to its ability to effectively leverage normal data mixed with anomalous data in the overall dataset. Excluding the extreme case of MTAD-GAT on the NAB dataset, F1 score drops at most by 0.04 when trained using NARCISSUS. Overall, NARCISSUS maintains comparable accuracy to semi-supervised methods (with their original training setup using normal data), achieving this with just unlabeled data.

### 5.3 BEYOND TIME SERIES DATA: ANOMALY DETECTION ON IMAGES AND GRAPHS

NARCISSUS can be generalized to AD with other types of data beyond time series. Conceptually, an image or graph can be treated as analogous to a time series segment. In anomaly detection for images and graphs, some tasks focus solely on a supervised approach with labeled data to detect predefined anomalies. Here, we instead focus on cases that rely on semi-supervised AD to demonstrate the ability of NARCISSUS to achieve equivalent performance with unsupervised training. Specifically, we consider three representative semi-supervised AD methods, as follows:

- PatchCore (Roth et al., 2022), a reconstruction based method for image AD.
- AnoGAN (Schlegl et al., 2017), an unconditional generation based method for image AD.
- AddGraph (Zheng et al., 2019b) for anomaly detection in dynamic graphs.

We train PatchCore with NARCISSUS on the MVTec2D dataset. We do the same for AnoGAN using the MNIST dataset (Schlegl et al., 2020). We use the official implementation of AddGraph (Zheng et al., 2019a) and train it following NARCISSUS approach using the UCI Message and Digg (a social news site) dataset. By default, we merge the original training and test data in these works and then apply NARCISSUS to detect anomaly in an unsupervised manner, without changing any other configuration. We apply NARCISSUS as well as the implementation approaches (ii) and (iii) in §5.1, where the bootstrapping is repeated 30 times randomly.

The performance is included in Table 3 for PatchCore, Table 4 for AnoGAN, and Table 5 for AddGraph, where we use the metrics as in those original works for performance evaluation. Training with NARCISSUS shows performance comparable to training on the entire normal dataset, with only negligible changes in metrics such as AUC. Achieving high-accuracy detection without labels is a non-trivial task. We also implemented a bootstrapping approach, where the training set is randomly sampled, and the model is trained to convergence. Bootstrapping resulted in significant performance degradation on PatchCore and AddGraph, while its performance on the MNIST dataset remained more stable, likely due to MNIST's synthetic nature and the sparse, pronounced anomalies.

### 5.4 ABLATION STUDY

For ablation study, we tried the unsupervised learning without VES or without RVES (i.e., only do VES in one shot).

Without VES, we can only train the model in a bootstrapping manner with random sampled training set. We conduct bootstrapping training with three representative methods: GDN (Deng & Hooi, 2021), TranAD (Tuli et al., 2022), and NPSR (Lai et al., 2024). We repeat 30 times randomly during

| Method | Dataset | Precision↑ | Recall↑ | Sensitivity↑ | Specificity↑ | AUC↑ |
|---|---|---|---|---|---|---|
| Semi-supervised AddGraph | UCI Message | 0.8054 | 0.6955 | 0.7082 | 0.8439 | 0.8050 |
| | Digg | 0.8282 | 0.7431 | 0.7434 | 0.8274 | 0.8470 |
| Unsup. AddGraph w/ bootstrapping | UCI Message | 0.6995 | 0.7034 | 0.7015 | 0.6634 | 0.7383 |
| | Digg | 0.8286 | 0.7021 | 0.6903 | 0.8192 | 0.8244 |
| Unsup. AddGraph w/ NARCISSUS | UCI Message | 0.8178 | 0.6899 | 0.7015 | 0.8496 | 0.8147 |
| | Digg | 0.8336 | 0.7404 | 0.7412 | 0.8301 | 0.8452 |

Table 5: AddGraph (Zheng et al., 2019b) evaluation using its official setup with a mixture of normal and anomalous data.

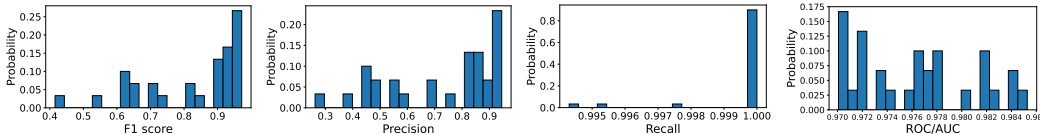

Figure 3: Bootstrapping performance with TranAD on MBA dataset.

bootstrapping. The detailed result is enclosed in §A.6, Table 6. In the worst case, bootstrapping may detect nothing when the majority of sampled data are anomalous.

Bootstrapping is unreliable due to the significant randomness in training data selection, making it impossible to determine which trained model performs better. For example, when applying bootstrapping to TranAD on the MBA dataset, which mixes normal and anomalous data, the performance is highly unstable, with F1 scores ranging from 0.43 to 0.97, as shown in Figure 3. Across all time series datasets, bootstrapping consistently exhibits unstable performance and, on average, performs significantly worse than NARCISSUS. More results in §A.6.

Without RVES in Alg. 2, the model may perform significantly worse because the validation set may not faithfully reflect the normal data. We present the statistics associated with RVES in §A.7, Table 7 with GDN, TranAD, and NPSR. NARCISSUS's final detection is jointly influenced by the models in the ensemble, thereby limiting the worst case performance.

## 6    DISCUSSION

**Limitation of NARCISSUS**. NARCISSUS can only be used when data is well bounded and anomaly is sparse. However, in real-world scenarios where a significant portion of the data may be anomalous, semi-supervised approaches trained with normal data may be necessary. Additionally, NARCISSUS requires a relatively large dataset to execute RVES effectively. After partitioning a validation set, the remaining data must be sufficient to train the base model. A notable exception was observed when applying NARCISSUS to MTAD-GAT on the NAB dataset, highlighting that NARCISSUS performs more reliably with larger datasets.

**Less image and graph cases are studied**. In image and graph anomaly detection, leveraging labeled data is common, with models trained to identify specific objects (Acsintoae et al., 2022; Tang et al., 2023). Moreover in models like PatchCore (Roth et al., 2022), all features are extracted using a pretrained model, limiting our ability to carry out more reliable anomaly detection. However, our experiments still demonstrate the potential of NARCISSUS in extending beyond time series to other domains.

## 7    CONCLUSIONS

In this paper, we have introduced NARCISSUS, a novel unsupervised learning approach that achieves accuracy on par with state-of-the-art semi-supervised methods but solely with unlabeled data. The success of NARCISSUS stems from our key insight: provided that anomalies are sparse and the data is well-bounded, the model training with mixed normal and anomalous data initially converges on the normal data. Comprehensive experiments demonstrate the effectiveness of the NARCISSUS approach and highlight its potential for AD with time series and other types of data.

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

# A APPENDIX

## A.1 APPLY EL2N SCORE ON ANOMALY DETECTION

**Lemma A.1.** *The training of arbitrary semi-supervised anomaly detection model $F$ trained on normal data $\mathcal{D}$ with L1 loss, L2 loss, and mixture, if the data in $\mathcal{D}$ is bounded by $\eta$, then it will meanwhile converge to a binary classification model trained with CE loss with measurable difference on loss.*

*Proof.* When the value is bounded by $\eta$, the target prediction can be normalized by $\frac{1}{\eta}$, so it is equivalent to train with softmax as the final activation. When a model trained with softmax and converge on L1 or L2 loss, these loss can be taken as approximation of CE loss according to the Taylor series expansion.

In Binary Cross-Entropy Loss, for a single example with true label, $y \in \{0, 1\}$, and predicted probability $\hat{y} \in (0, 1)$,

$$\mathcal{L}_{CE}(y, \hat{y}) = -[y \log(\hat{y}) + (1 - y) \log(1 - \hat{y})]$$

If we know the label is according to a fixed threshold, i.e.,

$$y = \begin{cases} 0, (y^* - \hat{y})^2 > \epsilon \\ 1, (y^* - \hat{y})^2 \leq \epsilon \end{cases}$$

The Taylor series expansion at $y^*$ is:

$$\begin{aligned}
\mathcal{L}_{CE}(y, \hat{y}) = &-y[\log(y^*) + \frac{1}{y^*}(\hat{y} - y^*) - \frac{1}{2y^{*2}}(\hat{y} - y^*)^2 + \cdots] \\
&- (1 - y)[\log(1 - y^*) - \frac{1}{1 - y^*}(\hat{y} - y^*) - \frac{1}{(1 - y^*)^2}(\hat{y} - y^*)^2 \cdots]
\end{aligned}$$

(4)

When a model converge on ground truth $y^*$ at arbitrary input, we have $\hat{y} \to y^*$, $|\hat{y} - y^*| \to 0$, and hence $y = 1$.

then we have

$$\mathcal{L}_{CE}(1, \hat{y}) = -y[\log(y^*) + \frac{1}{y^*}(\hat{y} - y^*) - \frac{1}{2y^{*2}}(\hat{y} - y^*)^2 + \cdots] \qquad (5)$$

when $y^* \to 1$,

$$\mathcal{L}_{CE}(1, \hat{y}) = -(\hat{y} - y^*) + \frac{1}{2}(\hat{y} - y^*)^2 + \cdots$$

For L2 loss, we have $\hat{y} \in (0, 1)$ and $y^* \in (0, 1)$,

$$\mathcal{L}_{\text{L2}}(y^*, \hat{y}) = \frac{1}{2}(y^* - \hat{y})^2$$

Therefore when $y^* \to 1$ (*i.e.*, the value of different data is very close), and $y$ is defined as aforementioned, then converge on $\mathcal{L}_{\text{L2}}$ is equivalent to converge on the CE Loss.

In more general case, we have

$$\begin{aligned}
|\mathcal{L}_{CE}(1, \hat{y})| &= |\log(y^*) - \frac{1}{y^*}(\hat{y} - y^*) + \frac{1}{2y^{*2}}(\hat{y} - y^*)^2 + \cdots| \\
&\approx |\log(y^*) - \frac{1}{y^*}(\hat{y} - y^*) + \frac{1}{2y^{*2}}(\hat{y} - y^*)^2| \\
&\approx |\log(y^*)| \\
&\geq \frac{1}{2}(y^* - \hat{y})^2
\end{aligned} \qquad (6)$$

because $y^*$ does not close to 1. $\qquad \square$

**Theorem A.2.** *The training dynamic with L1 or L2 loss cannot faithfully reflect the convergence on the binary classification task, with a constant difference* $\log y^*$*, which is agnostic to the model structure and other training schemes.*

*Proof.* Note that when model converge on L1 or L2, and if the value is not large, there is a **model agnostic** difference to the model trained with CE loss at $\log y^*$ in Equation 6. To avoid $\log 0$ issue we simply add $\delta$ to original data to avoid zeros, which does not influence prediction and reconstruction accuracy. $\qquad \square$

According to Theorem A.2, we can obtain the contribution of each sample on the anomaly detection (a binary classification task) via observing the L1 or L2 loss on regression task. From previous work (Paul et al., 2021), we know the samples that contribute less to binary classification task is stable at two metrics GraNd and EL2N. Here we focus on the EL2N score as it shows empirically better accuracy in (Paul et al., 2021). With L1/L2 loss, the EL2N score is estimated as:

$$\mathbb{E}||p(\mathbf{w}_t, x)|_{\text{CE}} - y||_2 = \mathbb{E}||p(\mathbf{w}_t, x)|_{\text{L2}} - y - \log y^*||_2 = \mathbb{E}||p(\mathbf{w}_t, x)|_{\text{L2}} - 1 - \log y^*||_2$$

Here we implicitly do two things: (1) when L2 does not converge, we do not compute its EL2N score, because obviously it is still valuable to a simpler task (*e.g.*,L2 regression); (2) when model easily converge on L2 loss, we can approximate its EL2N score as $y^* - 1 - \log y^* < y^* - 1 - \log \delta$, this points to the samples with smaller values. Let $\log \delta = -1$, then the EL2N score is bounded by $y^*$.

With the calibrated EL2N score, we can filter out a subset of data that contributes most to the anomaly detection problem. To have a high EL2N score, the sample must have significant value and not fit on the L2 loss. According to (Paul et al., 2021), this set should be stable from the early phase on training.

The result with calibrated EL2N (Cal-EL2N) score is shown in Figure. Where we find that Cal-EL2N fails to highlight informative (anomalous) part of data.

***Takeaway***: Essentially, this analysis shows that if two samples yield similar prediction (*i.e.*, $y^*$ is similar), then **their difference on L2 loss cannot faithfully reflect their difference on the EL2N**. Only in special case, for samples with large value or very small value, converge on L2 loss is a high order approximation of the CE loss, and hence L2 loss can directly show the difference on EL2N score in that case.

## A.2  PROOF OF LEMMA 4.1

Let $\mathbb{X}$ denote the set of all input data points, and let $\mathbb{Y} \subseteq \mathbb{X}$ be an arbitrary subset of $\mathbb{X}$. Suppose the function $f : \mathbb{X} \to \mathbb{Z}$ is a mapping defined over $\mathbb{X}$. The optimization problem defined in Eq. 3 can be formulated as:

$$\min_f \mathcal{L}(f, y) \quad \text{subject to} \quad \mathcal{C}(x, f(x), y) \leq \epsilon, \quad \forall x \in \mathbb{X},$$

where $\mathcal{L}(f, y)$ is the objective function, $\mathcal{C}(\cdot)$ represents the constraint function, and $\epsilon$ is a tolerance parameter.

Proof by contradiction: Suppose that the optimization problem defined in Eq. 3 has no solution. This would imply that the set of constraints cannot be satisfied simultaneously under the given objective function. Specifically, once the function $f$ and the corresponding input $y$ are known, the objective function becomes deterministic and thus computable, resulting in a feasible region of zero measure in the solution space. i.e.,

$$\mu(\{x \in \mathbb{X} | \mathcal{C}(x, f(x), y) \leq \epsilon\}) = 0,$$

where $\mu(\cdot)$ denotes the measure of the feasible region in the solution space.

Now, consider a subset $\mathbb{Y} \subset \mathbb{X}$ such that $|\mathbb{Y}| < N_{\max}$, where $N_{\max}$ denotes the maximum allowable size of a subset that satisfies the constraints. If for every point $x \in \mathbb{X} - \mathbb{Y}$, the model sequence $\{f_n\}$ converges, i.e.,:

$$\lim_{n \to \infty} f_n(x) = f(x),$$

and can be extended to converge on any subset $\mathbb{Y}$ with $|\mathbb{Y}| < N_{\max}$, then the convergence property of $f$ can be generalized to the entire domain $\mathbb{X}$. That is, there exists a function $\widehat{f}$ such that

$$\widehat{f} = f(x), \forall x \in (\mathbb{X} - \mathbb{Y}) \cup \mathbb{Y}$$

which satisfies all constraints defined.

This leads to the contradiction since the model $f$ can satisfy the constraints over every subset $\mathbb{Y} \subset \mathbb{X}$ with $|Y| < N_{\max}$ and the convergence can be generalized to the entire set $\mathbb{X}$.

## A.3  COMPLETE VERY EARLY STOPPING ALGORITHM

The Alg. 3 is the complete version of Alg. 1 in §4.4. Alg. 3 leverages the sparsity in the dataset, removing potential outliers in validation dataset in step 17 and 18. Therefore, the rest part of validation are very like to be normal data and faithfully reflect the convergence on normal dataset.

## A.4  REASONING OF VALIDATION SET SELECTION

Here we leverage the sparsity of anomalous data. Specifically, the following assumption is introduced on random selected validation set $\tau_i$,

$$p(|\tau_i \cap \mathbb{Y}| > 0) \approx p_{\text{GT}} \tag{7}$$

where $\sum \tau_i = \mathbb{X}, \tau_i \cap \tau_j = \varnothing, |\tau_i| = |\tau_j|, \forall i \neq j$, $p_{\text{GT}}$ is the ground truth sparsity of anomalous samples, $\tau_i$ represents a random sampled batch from time series (or images). To ensure that this assumption holds in practice, we limit the size (even) of the sampling set $|\tau_i| \ll |\mathbb{X}|$. We can easily verify this phenomenon by investigating all existing datasets with a mixture of normal and anomalous data.

Then with the definition of sample $\tau_i$, we select a subset $\mathbb{T}'$ of elements in $\mathbb{T} = \{\tau_i\}$ to perform as the validation set jointly. Due to the law of large numbers, by sampling sufficient number of $\tau_i$, *i.e.*, large $|\mathbb{T}'|$,

$$\tau_i \in \mathbb{T}', \ |\mathbb{T}'| > T \to p(|\tau_i \cap \mathbb{Y}| > 0) \approx p_{\text{GT}} \tag{8}$$

---

**Algorithm 3** Very Early Stopping for Unsupervised Anomaly Detection

---

**Require:** $|\mathbb{T}'| > T$
**Ensure:** $\mathcal{E} = E[(E(\tilde{f}) - E(f))|\mathbb{U}] < \epsilon,$
 1: $\mathbb{T}' \leftarrow$ random sampling by **masking** $\mathbb{T}$
 2: $\mathbb{V} \leftarrow \{v_i\}, v_i = 0 \forall i < |\mathbb{T}'|, i \in \mathbb{N}$        $\triangleright$ $\mathbb{V}$ is list of the mean loss on each validations
 3: Initialize $\eta, J$
 4: $N_{\max} \leftarrow n$                          $\triangleright$ $N$ is maximal number of epochs
 5: $M \leftarrow m$                            $\triangleright$ $M$ is minimal number of epochs
 6: $\epsilon \leftarrow 0 < x \ll 1$           $\triangleright$ Configure the convergence requirement of normal data
 7: $\delta \leftarrow 0 < x \ll 1$       $\triangleright$ Configure the convergence requirement of objective function
 8: $N \leftarrow 0$
 9: **while** $N \leq N_{\max}$ **do**
10:      **if** $N > M$ **then**
11:          $\mathcal{L}_{\mathbb{Y},N} \leftarrow \frac{\sum_{y \in \mathbb{Y}} \mathcal{L}(f(y_i), y_i)|_{y_i \in \mathbb{Y}}}{N_{\mathbb{Y}}}$          $\triangleright$ Compute the objective function, $\mathbb{Y}$ by $\eta\%$
12:          $i \leftarrow 0$
13:          **while** $i < \frac{|\mathbb{T}'|}{|\tau_i|}$ **do**
14:              $v_i \leftarrow \frac{\sum_{\forall t \in \tau_i} \mathcal{L}(t, f(t))}{|\tau_i|}$
15:              $i \leftarrow i + 1$
16:          $\mathbb{V}' \leftarrow \{v_i\}, v_i < q_{\text{mean}}(\mathbb{V}, \eta\%)$     $\triangleright$ Filter out top $\eta\%$ loss on validations $\{\tau\}$ by mean
17:          $\mathbb{V}^* \leftarrow \{v_i\}, v_i < q_{\max}(\mathbb{V}, \eta\%)$ $\triangleright$ Filter out top $\eta\%$ loss on validations $\{\tau\}$ by maximum
18:          $\mathcal{E}_N \leftarrow \frac{\sum \mathbb{V}' \cap \mathbb{V}^*}{|\mathbb{V}' \cap \mathbb{V}^*|}$                       $\triangleright$ Compute the constraint
19:          **if** $\mathcal{E}_N < \epsilon$ or $|\mathcal{E}_N - \mathcal{E}_{N-1}| < \alpha \cdot \epsilon$ **then**     $\triangleright$ converge or stop updating on validating set
20:              **if** $\exists j, |\mathcal{L}_{\mathbb{Y},N}| - |\mathcal{L}_{\mathbb{Y},N-j}| < 0, (j \in [1 \ldots, J])$ or $|\mathcal{L}_{\mathbb{Y},N} - \mathcal{L}_{\mathbb{Y},N-1}| < \delta$ **then**
21:                 **Break**
22:      $N \leftarrow N + 1$
23:      $\nabla \mathcal{L}(x, f(x)), \ x \in \mathbb{X} - \mathbb{T}', \mathbf{SGD}(f)$          $\triangleright$ Update the $f$ model on training set

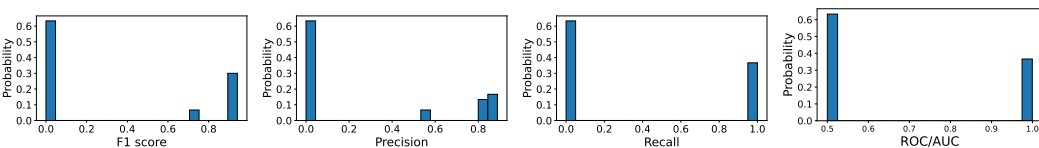

Figure 4: Bootstrapping performance with TranAD on NAB dataset.

## A.5 TRICKS TO ACCELERATE VES

We can apply the following techniques to accelerate the VES algorithm. While these optimizations have shown empirical success on the datasets we've tested, due to the inherent complexity of real-world scenarios, we recommend using the complete VES setup to ensure robustness.

- Only consider the validation set $\tau_i \in \mathbb{T}_i$ with minimal loss in VES to determine convergence level.
- Stop right after the $\tau_i$ with minimal loss converge, without check the objective function.
- Skip the RVES process if the validation sets exhibit distinct convergence speeds, or if all validation sets converge quickly and uniformly, similar to the majority of the training set.

## A.6 DETAILED EVALUATION OF BOOTSTRAPPING BASED TRAINING

We implement the same bootstrapping method as Livernoche et al. (2024); Han et al. (2022) on all the time series datasets. Specifically, we evaluate this method on the state-of-the-art methods NPSR (Lai et al., 2024) and TranAD (Tuli et al., 2022).

## A.7 DETAILED EVALUATION OF RANDOM SINGLE VES BASED TRAINING

We implement the VES as Algorithm 3 on all the time series datasets, **without taking the joint set**. Specifically, we evaluate this method on the state-of-the-art methods NPSR (Lai et al., 2024) and TranAD (Tuli et al., 2022). The result is included in Table 7.

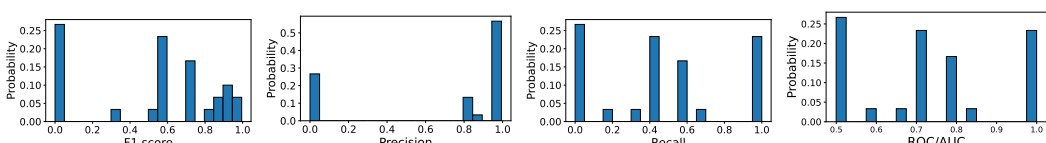

Figure 5: Bootstrapping performance with TranAD on SMAP dataset.

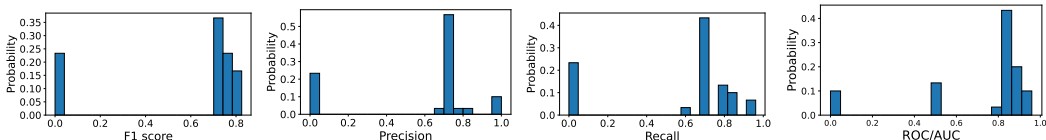

Figure 6: Bootstrapping performance with TranAD on SWaT dataset.

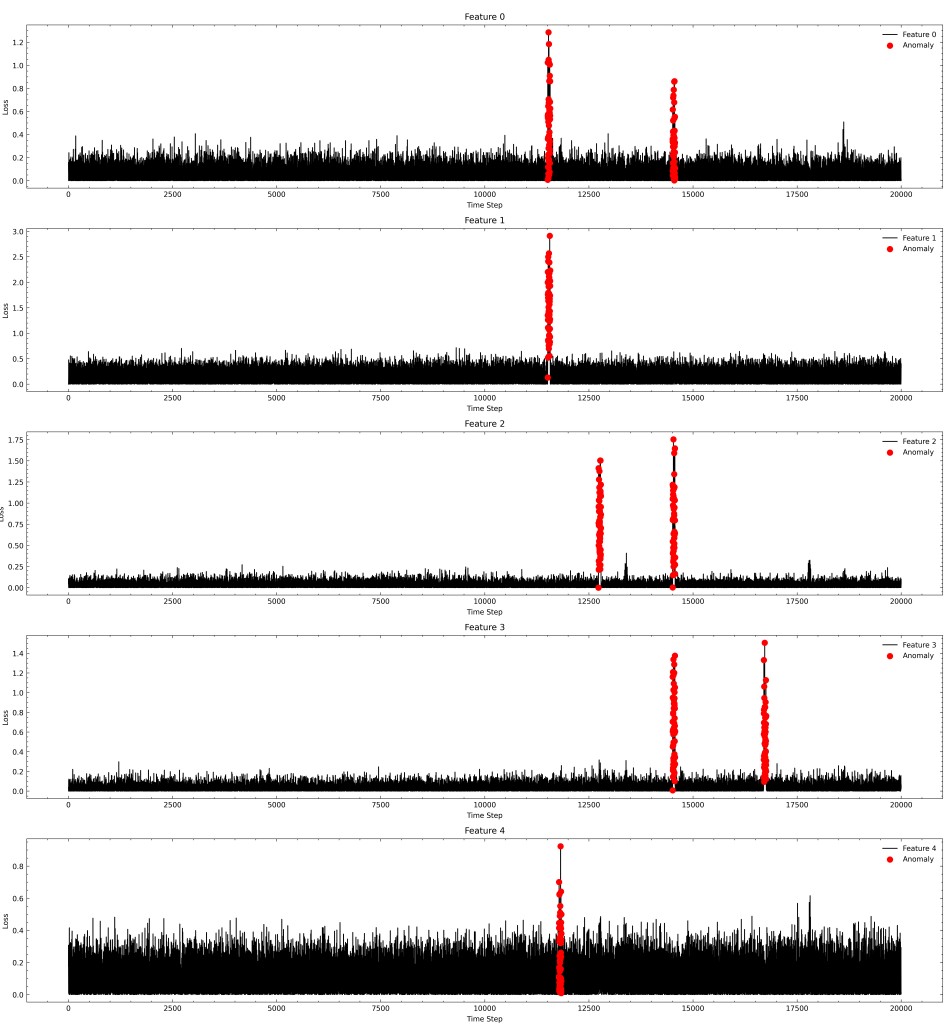

Figure 7: Unsupervised anomaly detection with Narcissus on a synthetic dataset, based method is TranAD

| Metrics | Stat. | NAB | | | MBA | | | SMD | | |
|---|---|---|---|---|---|---|---|---|---|---|
| | | GDN | TranAD | NPSR | GDN | TranAD | NPSR | GDN | TranAD | NPSR |
| P | Min | 0 | 0 | 0 | 0.3502 | 0.3273 | 0.4421 | 0 | 0. | 0 |
| | Max | 0.8889 | 0.8889 | 0.8571 | 0.8641 | 0.9486 | 0.8658 | 0.9883 | 0.9996 | 0.8242 |
| | Mean | 0.6112 | 0.6730 | 0.6281 | 0.5550 | 0.6373 | 0.6680 | 0.7303 | 0.8121 | 0.7962 |
| | NARCISSUS | 0.8889 | 0.8889 | 0.8571 | 0.8598 | 0.9461 | 0.8585 | 0.9814 | 0.9996 | 0.9117 |
| AUC | Min | 0 | 0 | 0 | 0.2954 | 0.3431 | 0.3562 | 0 | 0 | 0 |
| | Max | 0.9996 | 0.9996 | 0.9995 | 0.9597 | 0.9861 | 0.9603 | 0.8932 | 0.9955 | 0.9372 |
| | Mean | 0.9996 | 0.9996 | 0.9994 | 0.6570 | 0.7339 | 0.7538 | 0.7492 | 0.7841 | 0.6932 |
| | NARCISSUS | 0.9996 | 0.9996 | 0.9995 | 0.9583 | 0.9854 | 0.9582 | 0.8837 | 0.9220 | 0.9867 |
| F1 | Min | 0 | 0 | 0 | 0.3190 | 0.3626 | 0.3161 | 0 | 0 | 0 |
| | Max | 0.9412 | 0.9412 | 0.9231 | 0.9271 | 0.9736 | 0.9281 | 0.9394 | 0.9692 | 0.925 |
| | Mean | 0.4928 | 0.5328 | 0.6051 | 0.5281 | 0.6681 | 0.6249 | 0.5823 | 0.6320 | 0.6475 |
| | NARCISSUS | 0.9412 | 0.9412 | 0.9231 | 0.9246 | 0.9723 | 0.9244 | 0.9188 | 0.9152 | 0.8950 |

| Metrics | Stat. | SMAP | | | SWaT | | | Synthetic | | |
|---|---|---|---|---|---|---|---|---|---|---|
| | | GDN | TranAD | NPSR | GDN | TranAD | NPSR | GDN | TranAD | NPSR |
| P | Min | 0 | 0 | 0 | 0 | 0 | 0 | 0.4262 | 0.3187 | 0.4974 |
| | Max | 0.9440 | 0.9199 | 0.8676 | 1.0000 | 1.0000 | 1.000 | 0.9736 | 0.9776 | 0.9897 |
| | Mean | 0.5924 | 0.6192 | 0.6293 | 0.5261 | 0.6816 | 0.5793 | 0.5627 | 0.5638 | 0.5791 |
| | NARCISSUS | 0.9440 | 0.8503 | 0.9401 | 0.9762 | 0.9933 | 0.9977 | 0.9658 | 0.9776 | 0.9856 |
| AUC | Min | 0 | 0 | 0 | 0 | 0 | 0 | 0.4384 | 0.4916 | 0.4994 |
| | Max | 0.9823 | 0.9811 | 0.9835 | 0.8497 | 0.8466 | 0.8465 | 0.9993 | 0.9994 | 0.9997 |
| | Mean | 0.4322 | 0.4814 | 0.5174 | 0.4426 | 0.5429 | 0.5440 | 0.4919 | 0.4994 | 0.4995 |
| | NARCISSUS | 0.9823 | 0.9809 | 0.9820 | 0.8497 | 0.8436 | 0.8438 | 0.9991 | 0.9994 | 0.9996 |
| F1 | Min | 0 | 0 | 0 | 0 | 0 | 0 | 0.3763 | 0.3936 | 0.3859 |
| | Max | 0.9603 | 0.9199 | 0.9291 | 0.8166 | 0.8123 | 0.8116 | 0.9866 | 0.9887 | 0.9948 |
| | Mean | 0.4593 | 0.5126 | 0.5056 | 0.4189 | 0.5034 | 0.5088 | 0.4812 | 0.5224 | 0.5253 |
| | NARCISSUS | 0.9603 | 0.9191 | 0.9587 | 0.8166 | 0.8128 | 0.8143 | 0.9826 | 0.9887 | 0.9928 |

Table 6: Detailed performance of Bootstrapping: comparison of performance metrics – Precision (P), AUC, F1 – across various anomaly detection methods on multiple datasets.

| Metrics | Stat. | NAB | | | MBA | | | SMD | | |
|---|---|---|---|---|---|---|---|---|---|---|
| | | GDN | TranAD | NPSR | GDN | TranAD | NPSR | GDN | TranAD | NPSR |
| P | Min | 0.8571 | 0.8276 | 0.8000 | 0.8502 | 0.9279 | 0.8453 | 0.7887 | 0.9141 | 0.8006 |
| | Max | 0.8889 | 0.8889 | 0.8571 | 0.8641 | 0.9486 | 0.8658 | 0.8012 | 0.9996 | 0.8245 |
| | Mean | 0.8730 | 0.8730 | 0.8286 | 0.8550 | 0.9390 | 0.8600 | 0.7961 | 0.9320 | 0.8123 |
| | NARCISSUS | 0.8889 | 0.8889 | 0.8571 | 0.8598 | 0.9461 | 0.8595 | 0.7980 | 0.9996 | 0.8117 |
| AUC | Min | 0.9995 | 0.9994 | 0.9993 | 0.9549 | 0.9801 | 0.9531 | 0.9856 | 0.9212 | 0.9762 |
| | Max | 0.9996 | 0.9996 | 0.9995 | 0.9597 | 0.9861 | 0.9603 | 0.9900 | 0.9955 | 0.9887 |
| | Mean | 0.9996 | 0.9996 | 0.9994 | 0.9570 | 0.9838 | 0.9583 | 0.9876 | 0.9501 | 0.9358 |
| | NARCISSUS | 0.9996 | 0.9996 | 0.9995 | 0.9583 | 0.9854 | 0.9582 | 0.9872 | 0.9220 | 0.9867 |
| F1 | Min | 0.9231 | 0.9057 | 0.8889 | 0.9190 | 0.9626 | 0.9161 | 0.8310 | 0.9080 | 0.8875 |
| | Max | 0.9412 | 0.9412 | 0.9231 | 0.9271 | 0.9736 | 0.9281 | 0.8362 | 0.9692 | 0.8950 |
| | Mean | 0.9328 | 0.9328 | 0.9055 | 0.9226 | 0.9689 | 0.9244 | 0.8342 | 0.9320 | 0.8925 |
| | NARCISSUS | 0.9412 | 0.9412 | 0.9231 | 0.9246 | 0.9723 | 0.9244 | 0.8350 | 0.9152 | 0.8950 |

| Metrics | Stat. | SMAP | | | SWaT | | | Synthetic | | |
|---|---|---|---|---|---|---|---|---|---|---|
| | | GDN | TranAD | NPSR | GDN | TranAD | NPSR | GDN | TranAD | NPSR |
| P | Min | 0.9398 | 0.8392 | 0.7891 | 0.9634 | 0.9593 | 0.9512 | 0.9562 | 0.9486 | 0.9697 |
| | Max | 0.9440 | 0.9199 | 0.8676 | 1.0000 | 1.0000 | 1.000 | 0.9736 | 0.9776 | 0.9897 |
| | Mean | 0.9424 | 0.9162 | 0.8272 | 0.9861 | 0.9816 | 0.9794 | 0.9623 | 0.9638 | 0.9792 |
| | NARCISSUS | 0.9440 | 0.8503 | 0.9401 | 0.9762 | 0.9933 | 0.9977 | 0.9658 | 0.9776 | 0.9856 |
| AUC | Min | 0.9818 | 0.9792 | 0.9711 | 0.8385 | 0.8385 | 0.8385 | 0.9988 | 0.9986 | 0.9994 |
| | Max | 0.9823 | 0.9811 | 0.9835 | 0.8497 | 0.8466 | 0.8465 | 0.9993 | 0.9994 | 0.9997 |
| | Mean | 0.9822 | 0.9804 | 0.9777 | 0.8436 | 0.8439 | 0.8440 | 0.9989 | 0.9990 | 0.9995 |
| | NARCISSUS | 0.9823 | 0.9809 | 0.9820 | 0.8497 | 0.8436 | 0.8438 | 0.9991 | 0.9994 | 0.9996 |
| F1 | Min | 0.9581 | 0.9125 | 0.8821 | 0.8074 | 0.8065 | 0.8036 | 0.9776 | 0.9736 | 0.9846 |
| | Max | 0.9603 | 0.9199 | 0.9291 | 0.8166 | 0.8123 | 0.8116 | 0.9866 | 0.9887 | 0.9948 |
| | Mean | 0.9591 | 0.9162 | 0.9056 | 0.8109 | 0.8087 | 0.8088 | 0.9812 | 0.9812 | 0.9895 |
| | NARCISSUS | 0.9603 | 0.9191 | 0.9587 | 0.8166 | 0.8128 | 0.8143 | 0.9826 | 0.9887 | 0.9928 |

Table 7: Detailed performance of RVES: comparison of performance metrics – Precision (P), AUC, F1 – across various anomaly detection methods on multiple datasets.

