# OpenReview forum: "Narcissus: Leveraging Early Training Dynamics for Unsupervised Anomaly Detection"
_ICLR.cc/2025/Conference — ICLR 2025 Conference Withdrawn Submission_

### Official Review · Reviewer_KrnT · 2024-10-17

**Soundness:** 2
**Presentation:** 1
**Contribution:** 2
**Rating:** 5
**Confidence:** 3

**Summary:**

This paper proposes an unsupervised anomaly detection method that uses the training loss dynamics of the unlabeled data to get pseudo labels. Specifically, this paper found that the convergence speed of normal data is faster than that of anomalous data. This paper uses this observation to get pseudo labels. This approach is model-agnostic and thus can be applied to any semi-supervised anomaly detection methods. The experiments show the effectiveness of the proposed method.

**Strengths:**

- Using the training loss dynamics to exploit normal and anomalous data for anomaly detection is an interesting idea, even though a similar approach has been proposed in ordinary classification problems.
- The proposed method was evaluated using many datasets in various domains and evaluation metrics such as AUC, F1, etc.

**Weaknesses:**

- The presentation and clarity of this paper are not high. For example, in line 115, it is unclear whether $x_t$ is a scalar or a vector because the space it belongs is not specified. In line 140, there is no definition of $k$. In lines 191-192, $\\|$ is not necessary. In addition, the same symbol $t$ is used for the index of data and training iterations, which is confusing. Algorithms 1 and 2 are difficult to follow because many undefined words, such as $\varepsilon_N$ and validation sets, are in the main text.
- The proof of the theorem 4.2 is based on an upper bound for the norm of the loss. However, since the actual value may be smaller than the upper bound, does this argument make sense? (I can understand the argument if it uses a lower bound). In addition, did the authors investigate whether the assumptions about the gradient norm are valid in the experiments?
- There are unsupervised anomaly detection methods that can explicitly exclude anomalies in the unlabeled data in the training process, such as [a]. Since the motivations are the same, it is better to compare these methods to validate the approach's effectiveness.

[a] Qiu, Chen, et al. "Latent outlier exposure for anomaly detection with contaminated data." International conference on machine learning. PMLR, 2022.

**Questions:**

- In Table 2, why does the proposed approach that uses estimated normal labels outperform semi-supervised anomaly detection methods that use true normal data in some cases?
- What is the anomalous rate in datasets used for experiments? This is important information since the proposed approach uses the assumption of the low anomalous rate. In addition, it is better to investigate the robustness when changing the anomalous rates.
- Why are different methods (PatchCore and AnoGAN) used for different image datasets (MVTec2D and MNIST)?

---

### Official Review · Reviewer_yS1n · 2024-10-22

**Soundness:** 2
**Presentation:** 3
**Contribution:** 2
**Rating:** 3
**Confidence:** 4

**Summary:**

The paper proposed an add-in method for anomaly detection through the training loss based on the observation that the model converges first on normal data. The paper is easy to read and well-written.

**Strengths:**

1. The paper is well-written.
2. The idea is interesting.

**Weaknesses:**

The key insight of this paper is that when training a model for a learning task on a combination of normal and anomalous data, the model converges faster on normal data while struggling with anomalous data. However, this insight is not an original contribution of the paper.
The relationship between model training loss and data was first observed in the paper [1]. It explores the performance of deep neural networks (DNNs), including loss values, when trained on normal and noisy data. Anomalies can be considered a type of noise, so both can be seen as studies of the same phenomenon. Specifically, Section 3.2 of paper [1] proposes using loss values to compare normal and noisy data. In Section 4, paper [1] concludes through extensive experiments that real data examples are easier to fit than noise, similar to the current submission.

Subsequently, many new works have built on this finding to identify noise or anomalies in data, such as [2] and [3]. Paper [2] observes that noisy samples take longer to learn. Notably, Figure 1 in the paper [2] and Figure 1 in the current submission express almost the same conclusion and have a nearly identical presentation. Paper [3] introduces the concept of gradient norm to quantitatively describe the changes in loss on normal and noisy data. Therefore, this point cannot be considered a contribution of this paper.

Moreover, using ensemble methods for anomaly detection is also a common practice and lacks originality. For example, paper [4] proposes using embedding methods for anomaly detection. In summary, this paper lacks innovation in its contributions.
The current research is limited to describing and utilizing the phenomenon. It could enhance its contribution if the paper provides more theoretical explanations.

[1] A Closer Look at Memorization in Deep Networks.ICML.2017.

[2] Unsupervised Label Noise Modeling and Loss Correction.ICML.2019

[3] Gradient Harmonized Single-stage Detector.AAAI.2019.

[4] An embedding approach to anomaly detection.ICDE.2016

**Questions:**

What is the difference between the proposed work and the works mentioned in the weakness?

---

### Official Review · Reviewer_XkXJ · 2024-11-02

**Soundness:** 3
**Presentation:** 4
**Contribution:** 3
**Rating:** 5
**Confidence:** 3

**Summary:**

This paper aims to address how to perform unsupervised anomaly detection while ensuring high performance. The overall framework begins by viewing the task from a semi-supervised model, where the goal is to first train a model that can describe normal data using purely normal data. This pre-trained model is then used as a classifier, with the idea that a high reconstruction error indicates anomaly points. However, in an unsupervised scenario, we do not have labeled data, and thus, there is no pure normal data available. Consequently, the problem shifts to how to distinguish between normal and abnormal data within the dataset.

In this context, a phenomenon is observed: normal data allows the model to converge faster during the training process, while abnormal data contributes very little to the model's convergence. The authors aim to leverage this phenomenon to differentiate between abnormal data and to train the pre-trained model using only normal data.

To validate the effectiveness of their method, the authors employ Robust VES (RVES), which enhances robustness by applying the VES procedure with multiple random validation sets. They compare NARCISSUS against various baselines, including unsupervised and semi-supervised methods, and demonstrate that it significantly outperforms existing detection methods across key metrics like precision, recall, F1 score, and AUC on widely used datasets. This establishes NARCISSUS as a strong approach for effective anomaly detection in unsupervised settings.

**Strengths:**

### 1. Originality
The paper presents an original approach by modifying the pretraining step to address the unsupervised setting, based on an observed phenomenon that serves as a prior belief.

### 2. Quality
The problem definition is clear, and the methodology is technically sound.

### 3. Clarity
The writing is highly readable and well organized, making it easy to follow the authors' arguments.

### 4. Significance
The topic of this paper is quite significant, as it explores the direction of extracting prior beliefs from observed phenomena to enhance model performance, representing a promising approach in the field.

**Weaknesses:**

1. **Lack of Explanation for Observed Phenomenon**
The observation that normal data leads to faster model convergence is insightful, but the paper seems to lack an explanation for why this occurs. If possible, the authors could provide some theoretical analysis or empirical investigation—such as examining properties of normal vs. anomalous data or analyzing gradient dynamics—to clarify the mechanism and help define the scope and limitations of the prior belief derived from this phenomenon.

2. **Insufficient Persuasiveness in Experimental Section**
The experimental section lacks some essential details. Key information such as hyperparameter sensitivity analysis, specific settings, and the anomaly ratio in datasets is missing. These elements are crucial for validating the robustness and replicability of the proposed method, especially in an unsupervised learning context.
   - 2.1) Please provide a comprehensive list of hyperparameters used in the experiments, particularly noting whether the settings for baselines and the NARCISSUS model are identical when achieving optimal performance.
   - 2.2) Is there a hyperparameter sensitivity analysis to determine the importance of these parameters for model performance?
   - 2.3) Test whether different hyperparameter settings are needed to achieve optimal performance at varying anomaly ratios in the datasets.

**Questions:**

# Questions
If the authors can address some of the questions, it could enhance the overall rating of the paper. I encourage the authors to prioritize questions related to the experimental setup.
1. **Experimental Setup**
   - 1.1) The hyperparameters in the algorithm seem to significantly impact performance. Is there an analysis of hyperparameter sensitivity to model performance?
   - 1.2) Is the setting of hyperparameters influenced by the anomaly ratio in different datasets?
   - 1.3) When comparing with baselines, are hyperparameters kept consistent?

2. **Discussion of the Observed Phenomenon**
   The core idea of this paper is based on the observation that "the model converges faster on normal data while struggling to fit anomalous data."
   - 2.1) What might be the underlying cause of this phenomenon?
   - 2.2) Under what circumstances might this phenomenon fail to hold?

3. **Motivation**
   Overall, the performance of NARCISSUS relies heavily on its ability to identify abnormal data. Given this, I wonder why the authors chose not to use this method directly for anomaly detection rather than as a filtering step followed by pretraining. The two-stage approach (filtering then pretraining) seems to introduce added complexity, and a clearer rationale for this design choice would be helpful.
   - 3.1) Is it possible to design an algorithm to perform anomaly detection directly by relying on convergence speed during the training process?
   - 3.2) This comparison could clarify the advantages of NARCISSUS and reveal any trade-offs, such as differences in detection accuracy, computational efficiency, and robustness across various data distributions.
   - 3.3) An empirical evaluation of these alternatives could help illustrate cases where the two-stage method has clear benefits over a direct approach, providing a stronger justification for the current design.

---

### Official Review · Reviewer_pFCx · 2024-11-03

**Soundness:** 3
**Presentation:** 2
**Contribution:** 3
**Rating:** 5
**Confidence:** 3

**Summary:**

This paper presents NARCISSUS, which is a model-agnostic approach for unsupervised anomaly detection in time-series. Instead of relying on supervision signals, the proposed method excels in its capability of transforming any semi-supervised model into unsupervised with minimal adjustments, leveraging training dynamics to single out anomalous data without relying on labels or reconstruction error thresholds. Its key insight is simple but clever, that is, when training a model for some learning task on a mix of normal and anomalous data, the model converges faster on normal data while struggling to fit to anomalous data. Upon this insight, NARCISSUS uses early stopping to halt training before the model begins overfitting anomalies. The experiments are carried out over time series, image, and graph-based datasets, and empirical results show competitive or superior performance to semi-supervised baselines across various metrics, including AUC, F1 score, and precision.

**Strengths:**

+ The motivation and insight are interesting and legit.

+ The theoretical analysis has been done to quantify why early training dynamics can differentiate normal from anomalous data.

+ Empirical evidence is supportive and across multiple data modalities, although the model was originally tailored for time-series.

**Weaknesses:**

- The presentation needs to be improved. I struggle to follow its notation and derivations.

- While the ensemble approach improves robustness, the theoretical analysis is independent from it (why?).

- Due to the computational demanding nature of ensemble, benchmarks on training time, memory usage, and latency would be valuable additions to gauge the practical scalability of the proposal in large deployments.

- NARCISSUS assumes that anomalies are sparse and the data is well-bounded, which may limit its applicability to domains with high anomaly densities or unbounded data ranges (e.g., streams). How about the drift behaviors of anomalies which is also common in practice?

**Questions:**

I reproduce my weaknesses as questions to be addressed:

- The presentation needs to be improved. I struggle to follow its notation and derivations.

- While the ensemble approach improves robustness, the theoretical analysis is independent from it (why?).

- Due to the computational demanding nature of ensemble, benchmarks on training time, memory usage, and latency would be valuable additions to gauge the practical scalability of the proposal in large deployments.

- NARCISSUS assumes that anomalies are sparse and the data is well-bounded, which may limit its applicability to domains with high anomaly densities or unbounded data ranges (e.g., streams). How about the drift behaviors of anomalies which is also common in practice?

**Details Of Ethics Concerns:**

N/A, it's basic research.

---

### Note · Authors · 2024-12-05

**Comment:**

After careful consideration of the feedback and the current state of the paper, I have decided to withdraw it to undertake major revisions before resubmission.

Thanks

**Withdrawal Confirmation:**

I have read and agree with the venue's withdrawal policy on behalf of myself and my co-authors.